# Criteria-Based Approach to Select Relevant Environmental SDG Indicators for the Automobile Industry

**Sergej Lisowski [1,2,](ID), Markus Berger [2], Justus Caspers [2], Klaus Mayr-Rauch [1], Georg Bäuml [1] and Matthias Finkbeiner [2]**

[1] Department of Environmental Management Product, Volkswagen Aktiengesellschaft, Brieffach 011/1774, 38436 Wolfsburg, Germany; klaus.mayr-rauch@volkswagen.de (K.M.-R.); georg.baeuml@volkswagen.de (G.B.)

[2] Institute of Environmental Technology, Chair of Sustainable Engineering, Technische Universität Berlin, Office Z1, Strasse des 17. Juni 135, 10623 Berlin, Germany; markus.berger@tu-berlin.de (M.B.); j.caspers@tu-berlin.de (J.C.); matthias.finkbeiner@tu-berlin.de (M.F.)

[*] Correspondence: sergej.lisowski@volkswagen.de; Tel.: +49-5361-9-24750

**Abstract:** The United Nations Sustainable Development Goals (SDGs) cannot be met without the private sector. In order to contribute to the fulfillment of the SDGs, companies have to identify their influence and select relevant SDGs. However, so far no research has been conducted on the influence of companies or industries at the most concrete level in the SDG framework—the 247 SDG indicators. In this paper, a criteria-based approach to select relevant environmental SDG indicators for the automobile industry is developed. The three criteria—environmental impact, direct impact, and automobile impact—are defined. By means of a qualitative analysis, 31 influenceable indicators are selected and substantiated by an empirical analysis of the automobile industry's impact. These indicators belong to 12 SDGs and demonstrate the broad influence of the automobile industry. The outcome of this study is a structured procedure for selecting relevant environmental SDG indicators. This procedure can be applied by companies and can also be adapted to other economic sectors. Finally, it is possible to quantify the level of influence of the selected indicators and thus measure the contributions of companies or economic sectors to the fulfillment of the SDGs.

**Keywords:** Sustainable Development Goals (SDGs); 2030 agenda; relevant SDG indicators; environment; automobile industry; automotive industry; impact; influence; contribution; criteria

## 1. Introduction

For more than 30 years, mankind has been dealing with the questions of what sustainable development means in a globalized world and how it can be achieved. The United Nations (UN) plays a leading role in this task. Important milestones for sustainable development were:

- the so-called "Brundtland Report" [1] of 1987, with a definition for sustainable development;
- the Earth Summit in 1992, with the adoption of Agenda 21;
- the Millennium Development Goals (MDGs) of 2000 for poverty reduction [2].

In 2015 the UN eventually adopted the Sustainable Development Goals (SDGs) with the resolution "Transforming our world: The 2030 Agenda for Sustainable Development" [2]. The SDGs are a response to the current major sustainability challenges—such as poverty, inequality, and environmental degradation [3]. In 17 goals, with 169 targets (or sub-goals) and 247 indicators, the SDGs define and

measure sustainable development aspirations [4]. As Stafford-Smith et al. [5] point out, the "universal agreement on a defined set of goals and targets for global sustainability and human development is a remarkable achievement" (p. 912). The SDGs thus serve as a global platform or "guiding light" [3] (p. 176) for "Governments, the private sector, civil society, the United Nations system and other actors" to transform the world towards sustainable development [6] (p. 10).

After five years of SDGs—i.e., one-third of the time available to meet the goals by 2030—it can be concluded that the progress is insufficient. In other words, the world is "not on track" to meet the SDGs [7] (p. 35), [8] (p. 3) and [9] (p. 22). The performance of UN member states on the SDGs is tracked in the "SDG Index and Dashboards" of the Bertelsmann Stiftung and Sustainable Development Solutions Network. Even the top performing, North European countries in this index are not going to achieve all 17 SDGs. "Major performance gaps" were identified in environmental issues, such as climate change as well as biodiversity, on land and in water [9] (p. x). The Global Sustainable Development Report [10] reveals similar results: there is a great need for action on environmental issues in particular.

Even though the agenda calls different actors to step up, the private sector plays a crucial role in achieving the SDGs [6,11,12]. Experience with the MDGs has shown that the private sector must be closely involved if such goals are to be achieved successfully [12]. According to current surveys, many companies, especially large ones [13,14], are working with the SDGs [15]. However, implementation through dedicated SDG business goals has been slow—only about half of the companies surveyed have derived goals. Companies also need to deepen their efforts to understand and make full use of their influence. For example, only 37% of the companies surveyed consider impacts along the value chain. In addition, most companies examine their impacts at the overarching goal level, while only 36% analyze their impacts at the more detailed target level, as a joint survey by World Business Council for Sustainable Development and DNV-GL reveals [15]. In another survey this proportion is even less—in a case study for Colombia, only 11% of the companies in the sample analyzed their impacts on the target level [3].

One important part of the private sector is the automobile industry. Due to its global presence, the automobile industry can make a substantial contribution to sustainable development—for example, by offering modern mobility [16], economic growth [17], social security for a large number of employees [17], global influence via supply chains and production sites in almost 50 countries [18], as well as increasing sustainability awareness and action (such as electrification and decarbonization) [19,20]. At the same time, the automobile industry is also responsible for some environmental problems that hamper sustainable development. For example, in 2018 18% of global direct $CO_2$ emissions from fuel combustion were attributed to road transport [21]. In the European Union (EU), road transport is responsible for 19% of greenhouse gas emissions [22] and for 28% of air pollutant $NO_X$ [23] (both for 2017). In addition, for batteries for electric vehicles—which were developed to cut fuel combustion emissions—the use of resources that entail social and environmental issues in mining, such as cobalt and lithium, is currently essential [24,25].

The SDGs were designed by the UN as "integrated and indivisible" [6] (p. 1), so ideally companies should also contribute to all SDGs [26]. There are general guidelines available for implementing the SDGs in companies, such as SDG Compass [12] or the Chemical Sector SDG Roadmap [27]. When applying both guidelines, relevant SDGs should be selected, which is done by assessing the influence of the company. This approach is reasonable, as the impact on the SDGs differs between sectors and companies [14]. In order to select relevant SDGs or "prioritize" in the SDG Compass terms, concrete tools or selection criteria that can be applied by companies are currently missing. At the overarching goal level, this does not seem to be a problem with the 17 SDGs. However, due to the high number of 169 targets and 247 indicators [4], it can be concluded that companies need support in selecting relevant (i.e., influenceable in this context) targets and indicators [28]. This need becomes all the more important, as it is necessary to select relevant topics at these levels since only the indicators substantialize the content of the SDGs and make contributions measurable [3]. The need for methods to evaluate contributions to the SDGs is also described by Hák et al. [29].

The selection of relevant indicators reduces the variety and allows companies to work more effectively on fulfilling the SDGs. Some studies address the problem of selecting relevant SDGs or targets for industries or companies. In [30], van Zanten and van Tulder exclude and summarize SDG targets to reduce the total number to 59 and thereby select relevant targets for companies. For the automobile industry, the impact at the goal level was determined in [11] and [31] with different results. Furthermore, Betti et al. [32] developed an "Impact-Index" to measure the influence of sectors at the target level. For the automobile industry, these results are published aggregated at the overarching goal level. However, no analyses are yet available to determine the impact at the deepest level in the SDG structure, the indicator level. In addition, there are no published criteria for unbiasedly selecting indicators. However, these are needed to ensure that no "cherry-picking" [5] (p. 917) or "rainbow-washing" [26] (p. 254) is carried out while selecting relevant SDG indicators [30,33].

Therefore, the research questions of this paper are:

1.  Which criteria should be used to select relevant environmental SDG indicators?
2.  Which environmental SDG indicators are relevant for the automobile industry?

To answer these questions, the criteria for selecting relevant SDG indicators are defined in the chapter Materials and Methods. Using these criteria, relevant environmental SDG indicators for the automobile industry are subsequently selected from the latest SDG indicator set with 247 indicators [4]. For better understanding, additional examples of how this qualitative analysis was conducted are provided in this chapter, too. Additionally, empirical evidence for the impact of the automobile industry is examined for the selected indicators. The Results chapter contains the identified indicators and compares the results with other studies. The applied method, limitations, and results are discussed in the following chapter. In the final chapter of this paper, conclusions are shown and directions for future research are delineated. Additionally, an explanation of the complete indicator selection is provided in the Supplementary Materials.

## 2. Materials and Methods

The research questions of this paper are answered based on an analysis of the current global UN SDG indicators. The indicators examined are dated March 2020 and thus include the amendments of the 51st session of the UN Statistical Commission. This indicator set consists of 247 indicators in total. As some indicators are used multiple times to measure targets, 231 independent UN SDG indicators are listed [4]. Besides these global UN SDG indicators, there are also national (e.g., Germany [34]) and international (e.g., EU [35]) indicator sets derived from the UN member states or union of states [6] to contribute to the SDGs. What makes the global UN SDG indicators beneficial is the transnational availability of data, since all UN member states are obliged to report on them. In contrast, national or international indicators are linked to the 17 overarching goals, but may differ from the global UN SDG indicators (hereinafter termed just SDG indicators). Comparisons between countries are therefore not possible with these indicator sets. To select relevant environmental SDG indicators, criteria are first defined to characterize the term "relevant" in the sense of being influenceable [12,27]. For these criteria, it is then specified in which cases indicators are selected or not selected. Finally, the group of selected SDG indicators is sub-divided based on their ethical duties.

### 2.1. Criteria to Select Relevant Environmental SDG Indicators

Sustainability elements—economy, society, and ecology—are used in studies to categorize the SDGs [11,36]. The UN also developed the so-called "5Ps" (People, Planet, Prosperity, Peace, and Partnership) to which the SDGs contribute and into which they can be categorized [6]. In the theory of strong sustainability—for example, described by Shi et al. [37]—a sustainability hierarchy can be derived, which states that a healthy environment is the prerequisite for a functioning society and economy [36,38,39]. Due to this sustainability hierarchy and the increasing need for more environmental protection [9,10], environmental impact is determined as the first criterion. Ecosystems

including species, resources, and human health (as it is the purpose of environmental protection to preserve human health) are used as objects of protection, analogously to Hofstetter and Scheringer [40] and Mettier [41]. In order to make this protection comprehensive, both environmental conditions and impacts are considered.

Environmental impacts, as a result of environmental burdens, can be classified in different ways. The following types of environmental impact can be distinguished: direct and indirect, secondary, cumulative, synergistic, short, medium and long-term, permanent and temporary, reversible and irreversible, and positive and negative [42,43]. In this paper, environmental impacts are classified as either direct or indirect. The directness of impacts (or direct impact) is used as the second criterion, since such impacts are closely linked to the causative action [44]. The influence (of companies) is therefore more apparent for these than for other types of environmental impact.

The third criterion is the influence of the automobile industry (or automobile impact) on the SDG indicators. To determine the influence, the inputs and outputs of the main product of this industry, the automobile, are used. The system boundaries examined consider the entire life cycle—i.e., the extraction of raw materials, production (from parts, via components to the final product), use, and the end-of-life phase—of an automobile, analogously to Broch et al. [45] and Helmers et al. [46]. The inputs include all raw materials (such as metals, plastics, and renewable resources), energy, land, and water. Waste as well as emissions to air, water, and land are considered as outputs. Inputs and outputs were collected from two studies carried out by Joshi [47] and Dietz et al. [48].

The three criteria are shown in Figure 1.

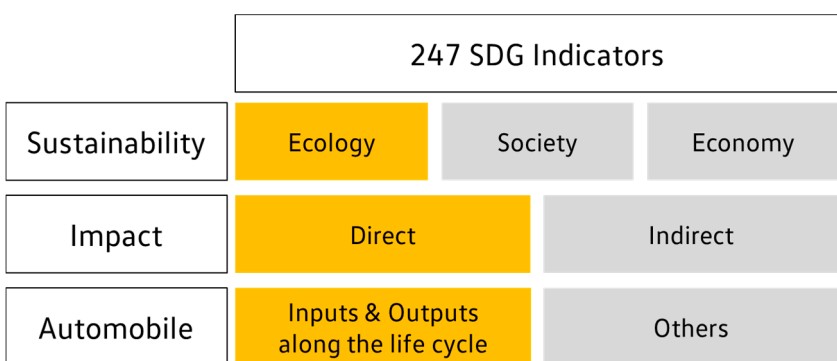

**Figure 1.** Overview of the 3 criteria to select relevant environmental SDG indicators (in yellow); the not-selected options per step are displayed in grey.

*2.2. Criteria-Based Selection of Relevant Environmental Indicators*

The basis for the selection of relevant environmental indicators is the available metadata, which contain the definition and calculation methodology of the SDG indicators [49]. At the time of this research (May 2020), provisional metadata were available for the latest indicator changes proposed to the 51st session of the UN Statistical Commission [50]. In addition, the methodology for some indicators is still under development. They are classified as "Tier III Indicators" [51]. For the three criteria, cases in which indicators are selected or not selected are defined. Definitions of the criteria and selection examples are presented in Table 1.

**Table 1.** Definitions of the criteria and selection examples.

| Step | Criterion | Selected When | Not Selected When | Example Indicators |
|---|---|---|---|---|
| 1 | Environmental impact | A change in the indicator level influences at least one of the environmental objects of protection: human health, ecosystem, and resources. | Indicator clearly has a social and/or economic content. | Selected: 13.2.2 Total greenhouse gas emissions per year. Not selected: 1.b.1 pro-poor public social spending. |
| 2 | Direct impact | Indicator measures environmental conditions (as change in an environmental condition directly impacts environment). or Indicator measures environmental impacts that directly lead to a change in environmental conditions (directly means that no other action is needed). Additionally: indicators with health impact are only assessed when they are caused by another environmental object of protection. | Indicator measures environmental impacts, that only indirectly lead to change in environmental conditions (indirectly means that other action is needed, the indicator works preparatory for a change in environmental conditions). Additionally: indicators with health impact are not caused by another environmental object of protection. | Selected (condition): 15.5.1 Red List Index. Selected (direct health impact): 3.9.1 mortality rate attributed to household and ambient air pollution. Not selected (indirect impact): 12.1.1 number of countries [ ... ] implementing policy instruments aimed at supporting [ ... ] sustainable consumption and production. Not selected (health without another environmental cause): 3.a.1 age-standardized prevalence of current tobacco use among persons aged 15 years and older. |
| 3 | Automobile impact | The indicator level is influenced by inputs or outputs of an automobile (along the life cycle). | No inputs or outputs of an automobile (along the life cycle) influence the indicator. | Selected: 14.1.1 (a) index of coastal eutrophication and (b) floating plastic debris density. Not selected: 7.b.1 installed renewable energy-generating capacity in developing countries [ ... ]. |

The selection is carried out in the form of a qualitative analysis in three steps. In each step, an evaluation is performed to determine whether the content of the indicators from the metadata meets the cases "selected" or "not selected" from Table 1. In the case of Tier III indicators, the evaluation is carried out on the basis of the designation. For an indicator to be selected, one example that meets the requirements is sufficient. In the first step, all 247 indicators are evaluated. In the following two steps, only those selected in the preliminary stage are evaluated.

*2.3. Structuring of the Relevant Environmental SDG Indicators*

In some studies, SDGs [11] or their targets [30] are distinguished on the basis of their ethical duties to either create positive value or to reduce harm. In the case of positive impacts, also called "doing good ", actions are not expected—i.e., they are perceived as additional. In contrast, for negative impacts, called "avoiding harm", it is expected that those causing problems will also solve them [52]. Hence, reducing harm can be seen as a stronger norm than creating positive value [30]. The two cases are illustrated by examples. If a company carries out activities to prevent its own emission of pollutants into the environment, it reduces negative impacts, i.e., avoiding harm. If, on the other hand, a company participates in water protection without polluting it, then it creates positive impacts, i.e., doing good. This distinction is used in this paper to structure indicators into positive and negative impacts. Therefore, the definitions of [30] are used:

- doing good: generate "positive externalities";
- avoiding harm: "reduce negative externalities".

For each relevant environmental SDG indicator, it is examined whether it corresponds to the positive or negative side. In addition, the content of its target is also examined in order to identify the purpose of the indicator.

*2.4. Empirical Evidence of the Impact and Comparison of the Results with other Studies*

For the selected indicators, the empirical evidence of the impact of the automobile industry was additionally examined. For this purpose, the calculation methodology of the indicators in the metadata [49] was first identified. Subsequently, the factors that cause a change in the indicators (i.e., for emissions or resource use) were analyzed. Finally, sector data (for the automobile industry or the transport sector) were searched for these factors. The study was conducted for the European region. Where available, data from the EU28 were used. Alternatively, data from Europe in general or from individual EU28 countries were used. The most recent data were selected. For each selected indicator, at least one impact example was examined, but no comprehensive analysis of all impacts was carried out. If no dedicated data were available for an indicator, a reasoned estimate was made based on existing data. It was not possible to determine an impact for Tier III indicators, as no agreed methodology is yet available for these indicators. As a result, the proportion of the automobile industry or of the transport sector in the factor that causes an indicator change is given. For example, the proportion of the transport sector in certain emissions or of the automobile industry's share in the use of resources is analyzed.

Furthermore, the selection results of the present paper were compared with the results of other studies. This comparison of the selection decisions for the three criteria takes place on the overarching goal level, since other studies provide results on this level only (with the exception of environmental indicators defined by the United Nations Environment Programme (UNEP) [53]). One comparison each was made for the categorization of SDGs to the environment (step 1 and 2 in Table 1), and for the impact of the automobile or transport sector on the SDGs (step 3 in Table 1). The comparison of the categorization of SDGs to the environment was carried out with the work of UNEP [53], Beck and Buddemeier [11], and the Stockholm Resilience Centre (SRC) [36]. For the results of the present paper, an SDG was attributed to the environment if it contains at least one of the selected indicators with an environmental impact (124 in step 1; 45 in step 2). In the case of UNEP, the same procedure was applied: if an SDG contains at least one of the 93 environmental indicators, it was attributed to the environment. For Beck and Buddemeier, the SDGs are presented which are attributed to the environment, while for the SRC those which are attributed to the biosphere are presented. The comparison of the impact of the automobile or transport sector on the SDGs was carried out with the Sector SDG Target Impact Index (SSTII) presented by Betti et al. [32], Beck and Buddemeier [11], and the World Benchmarking Alliance (WBA) [31]. For the results of the present paper, an SDG was considered influenceable if it contains at least one of the selected indicators with an automobile impact (31 in step 3). For the SSTII, the seven SDGs with the highest impact index value for the transportation sector were chosen for comparison. For Beck and Buddemeier, SDGs were chosen for comparison which have a high impact on at least two out of three value chain steps for the automobile sector, while, for the WBA, SDGs with a high impact for the automobile and components industry were chosen.

## 3. Results

Environmental impact, direct impact, and automobile impact were identified as criteria for selecting relevant environmental SDG indicators. Applying these criteria results in a stepwise reduction in the SDG indicators. This is illustrated in Figure 2. Out of 247 indicators, 124 have an environmental impact. Of these, 45 indicators have a direct impact on the environment and 31 indicators—27 of which are independent—can be influenced by the automobile industry. Within this group of relevant environmental indicators, 28 deal with the negative (avoiding harm) side of ethical duty, while three indicators cover the positive (doing good) side. The selection is illustrated by three examples in Table 2.

The evaluation of all 247 indicators is attached in Table S1 in the Supplementary Materials. Each selection decision is also explained there. In the following, the environmental indicators relevant to the automobile industry are separately presented.

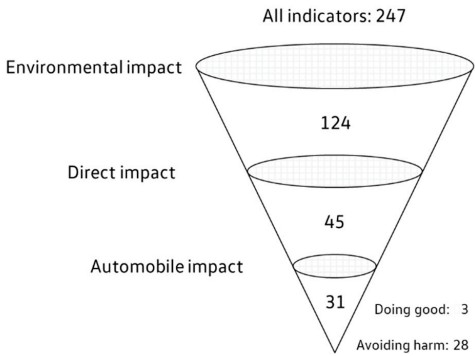

**Figure 2.** Stepwise reduction in the SDG indicators by each criterion.

**Table 2.** Examples for selecting relevant environmental SDG indicators for the automobile industry.

| | Indicator | 1. Environmental Impact | 2. Direct Impact | 3. Automobile Impact |
|---|---|---|---|---|
| 1.5.1 | Number of deaths, missing persons, and directly affected persons attributed to disasters [ … ] | Health | Impact Environmentally caused Disasters (e.g., climate change, extreme temperature, and heat waves [54]) directly [55,56] impact health (as indicator includes death, injury, illness, as well as other health effects). | Greenhouse gas emissions (like $CO_2$) cause climate change and thus extreme temperature and heat waves and are emitted by fuel combustion in the use phase and along automobile supply chains (output). |
| 11.6.2 | Annual mean levels of fine particulate matter (e.g., PM2.5 and PM10) in cities [ … ] | Health | Impact Environmentally caused Fine particulate matter directly impacts health. | Levels of fine particulate matter are influenced by emission of PM2.5 and PM10. These are mainly emitted by fuel combustion and the abrasion of tires and brakes in the use phase (outputs). |
| 15.5.1 | Red List Index | Ecosystem (species) | Condition The Red List Index measures the extinction risk of species and thus directly impacts the quality of ecosystems. | Species are threatened by habitat destruction, pollution, and climate change. Land use (a) [57], nitrogen (b) [58,59] and $CO_2$ (c) influence habitats, pollution, and climate change and (a) are caused by the production of automobiles and streets for the use of automobiles (input) as well as (b and c) being emitted by fuel combustion in the use phase and along automobile supply chains (output). |

### 3.1. Avoiding Harm Indicators

The 28 avoiding harm indicators and empirical evidence of the impact of the automobile industry are shown in Table 3. The impacts are described in more detail in Table S2 [60–96] in the Supplementary Materials. In addition, the calculation methodology for each indicator is also given there. It can be seen that these indicators are distributed over eleven SDGs. SDG 6 (clean water and sanitation) and SDG 15 (life on land), each with five indicators, contain the most, followed by SDG 12 (responsible consumption and production) with four indicators. In addition, three indicators occur multiple times. The indicators "Material footprint, material footprint per capita, and material footprint per GDP" and "Domestic material consumption, domestic material consumption per capita, and domestic material consumption per GDP" are used in both SDG 8 and SDG 12 to measure the targets. The indicator "Number of deaths, missing persons and directly affected persons attributed to disasters per 100,000 population" is used in SDG 1, SDG 11, and SDG 13 to measure their targets. Thus, 24 independent, environmental avoiding harm indicators relevant to the automobile industry were identified.

**Table 3.** Relevant environmental avoiding harm indicators for the automobile industry with impact examples for Europe.

| SDG | | Indicator | Impact Example for Europe |
|---|---|---|---|
| 1 | 1.5.1 | Number of deaths, missing persons, and directly affected persons attributed to disasters per 100,000 population. | Extreme temperatures are a consequence of climate change, which is directly caused by greenhouse gas (GHG) emissions. Extreme temperatures account for 0.2% of people affected by all disasters [60] and for 24.8% of the dead or missing people from all disasters [61]. Road transportation accounts for 20.9% of GHG emissions by fuel combustion for energy generation [62]. |
| 3 | 3.9.1 | Mortality rate attributed to household and ambient air pollution. | Road transportation accounts for 10.6% of all PM2.5 emissions [63]. |
| | 3.9.3 | Mortality rate attributed to unintentional poisoning. | No empirical impact could be found. |
| 6 | 6.3.1 | Proportion of domestic and industrial wastewater flows safely treated. | Motor vehicles and transport equipment account for a part of the generated wastewater. For all industries, 64.9% of wastewater is discharged after treatment [72]. |
| | 6.3.2 | Proportion of bodies of water with good ambient water quality. | Wastewater from industry can cause a deterioration of the water-quality [74]. For the proportion of treated industrial wastewater, see 6.3.1. |
| | 6.4.1 | Change in water-use efficiency over time. | Manufacturing industry accounts for 11.2% of the total water abstraction of fresh surface and groundwater [75]. Manufacture of transport equipment accounts for 2.5% of the total water used in the manufacturing industry [76]. |
| | 6.4.2 | Level of water stress: freshwater withdrawal as a proportion of available freshwater resources. | The indicator can be impacted by water withdrawal. For the proportion, see 6.4.1. |
| | 6.6.1 | Change in the extent of water-related ecosystems over time. | For the 1. sub indicator (spatial extent of water-related ecosystems), transport accounts for 1.1% of the net land take of wetlands, as well as for 0.4% of the net land take of water bodies [78]. |
| 7 | 7.3.1 | Energy intensity measured in terms of primary energy and GDP. | Transportation accounts for 28.5% of the total final energy consumption [79]. Meanwhile, the automobile industry accounts for over 7% of the EU GDP [80]. |
| 8 | 8.4.1 | Material footprint, material footprint per capita, and material footprint per GDP. | Direct impact on recourses that is influenced by material use in automobile production. Motor vehicles, trailers, and semi-trailers account for 2.4% of the total raw material flow [82]. |
| | 8.4.2 | Domestic material consumption, domestic material consumption per capita, and domestic material consumption per GDP. | For the proportion, see 8.4.1. |
| 9 | 9.4.1 | $CO_2$ emission per unit of value added. | Manufacturing industries and construction accounts for 15.7% of the total $CO_2$ emissions from fuel combustion [62]. The automobile industry accounts for about one third of the manufacturing industries and construction GDP [80,83]. The MVA of the automobile industry is about 4.7% the total EU GDP [80,83,84]. |
| 11 | 11.3.1 | Ratio of land consumption rate to population growth rate. | Transport accounts for 9.3% of the total land take [78]. |
| | 11.5.1 | Number of deaths, missing persons, and directly affected persons attributed to disasters per 100,000 population. | Same as 1.5.1. |
| | 11.6.2 | Annual mean levels of fine particulate matter (e.g., PM2.5 and PM10) in cities (population weighted). | Same as 3.9.1 for PM2.5. Road transportation accounts for 10.6% of PM2.5 emissions and for 10.2% of PM10 emissions [63]. |
| 12 | 12.2.1 | Material footprint, material footprint per capita, and material footprint per GDP. | Same as 8.4.1. |
| | 12.2.2 | Domestic material consumption, domestic material consumption per capita, and domestic material consumption per GDP. | Same as 8.4.2. |
| | 12.4.2 | (a) Hazardous waste generated per capita and (b) the proportion of hazardous waste treated, by type of treatment. | Discarded vehicles account for 6.3% of the total hazardous waste generation [85]. |
| | 12.5.1 | National recycling rate, tons of material recycled. | Calculation of impact is not possible, as the indicator methodology is not yet defined (Tier III). |
| 13 | 13.1.1 | Number of deaths, missing persons, and directly affected persons attributed to disasters per 100,000 population. | Same as 1.5.1. |

**Table 3.** *Cont.*

| SDG | | Indicator | Impact Example for Europe |
|---|---|---|---|
| | 13.2.2 | Total greenhouse gas emissions per year. | Road transportation accounts for 20.9% of the total GHG emissions [62]. |
| 14 | 14.1.1 | (a) Index of coastal eutrophication and (b) floating plastic debris density. | (a) Eutrophication is caused by nutrients, particularly phosphorus and nitrogen [86]. Road transportation accounts for 38.9% of $NO_X$ emissions and 1.3% of $NH_3$ emissions [63].<br>(b) In Germany, 74% of all plastic emissions in the environment are micro plastic. Tire abrasion from passenger cars accounts for 34.7% of all micro plastic emissions [87]. |
| | 14.3.1 | Average marine acidity (pH) measured at agreed suite of representative sampling stations, | Road transportation accounts for 28.3% of $CO_2$ emissions from fuel combustion [62]. |
| 15 | 15.1.1 | Forest area as a proportion of the total land area, | Transport accounts for 11.3% of the net land take of forests between 2000 and 2018 [78]. |
| | 15.2.1 | Progress towards sustainable forest management. | For the (1) sub-indicator (forest area annual net change rate), see same as 15.1.1.<br>For the (2) sub-indicator (above-ground biomass stock in forest),<br>automobiles and automobile components account for 1.5% of the total wood used [90]. |
| | 15.3.1 | Proportion of land that is degraded over the total land area. | For the (1) sub-indictor (trends in land cover), see 15.1.1 for the forest area.<br>For the (3) sub-indicator (carbon Stocks), see 15.2.1, as carbon stock is proportional to biomass [92]. |
| | 15.4.2 | Mountain Green Cover Index. | Trees that are used in the automobile industry also grow in mountains—i.e., Cork at altitudes of 1200 m (Schütt et al., cited after [93]). Thus the economic share of wood use (see 15.2.1) can be taken as an estimate. |
| | 15.5.1 | Red List Index. | Urban and transport infrastructure spread leads to landscape fragmentation [94] and thus to habitat destruction and degradation [57]. Transport accounts for 2.4% of the total land use [95] and the transport infrastructure accounts for 1.7% of the total land cover [96]. The increasing land take for transport shows that these areas continue to expand. For the proportion, see 15.1.1 for forest area.<br>Nitrogen emissions lead to eutrophication of ecosystems and thus affect biodiversity [58,59]. For the proportion, see 14.1.1 (a).<br>Climate change is changing the living conditions for species, causing their habitats to shift. For the proportion, see 13.2.2. |

### 3.2. Doing Good Indicators

The three environmental doing good indicators and empirical evidence of the impact of the automobile industry are listed in Table 4. More detailed descriptions of the impacts as well as the calculation methodology for each indicator are given in Table S3 [97–104] in the Supplementary Materials. Each indicator belongs to different SDGs and, except for indicator 2.4.1, comes from SDGs that have already been listed in the avoiding harm indicators.

**Table 4.** Relevant environmental doing good indicators for the automobile industry with impact examples for Europe.

| SDG | | Indicator | Impact Example for Europe |
|---|---|---|---|
| 2 | 2.4.1 | Proportion of agricultural area under productive and sustainable agriculture. | 2.5% (4.6 million ha) of the agricultural area is devoted to energy and biomass production [97]. In Germany, for example, 41% of the area devoted to energy plants is used for bioethanol and biodiesel/plant oil [98]. Bioethanol and biodiesel are added proportionally to the total fuel, such as E5 or E10 gasoline or B7 diesel in the EU [99]. |
| 7 | 7.2.1 | Renewable energy share in the total final energy consumption. | The share of renewable energy in the gross final energy consumption in the transport sector is 8% [100]. Road transport accounts for 72.7% of the total final energy consumption in the transport sector [101]. |
| 9 | 9.1.2 | Passenger and freight volumes, by mode of transport. | Transport equipment accounts for 4.1% of the total road freight transport [102], for 3.4% of the total goods transported by rail [103], and for 0.4% of the total goods transported on inland waterways [104]. |

### 3.3. Comparison of the Results with other Studies

A total of 31 environmental SDG indicators relevant to the automobile industry are identified, 27 of which are independent. These indicators belong to 12 SDGs and thus cover more than two thirds of the 17 SDGs. The results of the present paper are compared with other studies. This comparison of the selection decisions of the three criteria takes place on the overarching goal level, since, with the exception of environmental indicators defined by the UNEP [53], other studies provide results on this level only. First, the categorization of SDGs to the environment is shown in Table 5, including the interim results of the first two steps from the methods in the present paper.

**Table 5.** Overview of the SDGs [2] that are attributed to the environment. The two upper rows contain the source and information on the evaluation. The following rows show the relationship between the SDGs and the environment. Colored: SDG is attributed to the environment; white: SDG is not attributed to the environment.

| # | SDG | Present Paper — Indicator Level Step 1 * | Present Paper — Indicator Level Step 2 ** | UNEP [53] — Indicator Level *** | Beck and Buddemeier [11] — Goal Level **** | SRC [36] — Goal Level ***** |
|---|-----|:--:|:--:|:--:|:--:|:--:|
| 1 | No Poverty | ▮ | | ▮ | | |
| 2 | Zero hunger | ▮ | ▮ | ▮ | | |
| 3 | Good health and well-being | ▮ | ▮ | ▮ | | |
| 4 | Quality education | ▮ | | ▮ | | |
| 5 | Gender equality | ▮ | | ▮ | | |
| 6 | Clean water and sanitation | ▮ | ▮ | ▮ | ▮ | ▮ |
| 7 | Affordable and clean energy | ▮ | ▮ | ▮ | ▮ | |
| 8 | Decent work and economic growth | ▮ | ▮ | ▮ | | |
| 9 | Industry, innovation, and infrastructure | ▮ | ▮ | ▮ | | |
| 10 | Reduced inequalities | ▮ | ▮ | | ▮ | |
| 11 | Sustainable cities and communities | ▮ | ▮ | ▮ | ▮ | |
| 12 | Responsible consumption and production | ▮ | ▮ | ▮ | ▮ | |
| 13 | Climate action | ▮ | ▮ | ▮ | ▮ | ▮ |
| 14 | Life below water | ▮ | ▮ | ▮ | ▮ | ▮ |
| 15 | Life on land | ▮ | ▮ | ▮ | ▮ | ▮ |
| 16 | Peace, justice, and strong institutions | ▮ | | ▮ | | |
| 17 | Partnerships for the goals | ▮ | | ▮ | | |

Note: * SDGs containing at least one of the 124 indicators selected in methods step 1 (environmental impact); ** SDGs containing at least one of the 45 indicators selected in methods step 2 (direct environmental impact); *** SDGs containing at least one of the 93 indicators attributed to the environment; **** SDGs that are attributed to the environment; ***** SDGs that are attributed to the biosphere.

Table 6 shows an overview of the SDGs influenced by the automobile industry and transport sector, respectively, including the results of the third step from the methods in the present paper.

**Table 6.** Overview of the SDGs [2] influenced by the automobile industry and the transportation sector, respectively. The two upper rows contain the source and information on the evaluation. The following rows show the relationship between the SDGs and the automobile industry or the transportation sector. Colored: influence exists; white: no influence exists.

| SUSTAINABLE DEVELOPMENT GOALS | Present Paper — Indicator Level Step 3 * | SSTII [32] — Target Level ** | Beck and Buddemeier [11] — Goal Level *** | WBA [31] — Goal Level **** |
|---|---|---|---|---|
| 1 No Poverty | ● colored | | | |
| 2 Zero hunger | ● colored | | | |
| 3 Good health and well-being | ● colored | ● colored | ● colored | ● colored |
| 4 Quality education | | | | |
| 5 Gender equality | | ● colored | | |
| 6 Clean water and sanitation | | | ● colored | ● colored |
| 7 Affordable and clean energy | ● colored | ● colored | ● colored | ● colored |
| 8 Decent work and economic growth | ● colored | ● colored | ● colored | |
| 9 Industry, innovation, and infrastructure | ● colored | ● colored | ● colored | |
| 10 Reduced inequalities | ● colored | | | |
| 11 Sustainable cities and communities | ● colored | | | ● colored |
| 12 Responsible consumption and production | ● colored | ● colored | ● colored | ● colored |
| 13 Climate action | ● colored | ● colored | | ● colored |
| 14 Life below water | ● colored | | | |
| 15 Life on land | ● colored | ● colored | | |
| 16 Peace, justice, and strong institutions | ● colored | | | |
| 17 Partnerships for the goals | | | | ● colored |

Note: * SDGs containing at least one of the 31 indicators selected in methods step 3 (automobile impact); ** SDGs with the highest impact index values for the transportation sector; *** SDGs with a high impact in at least 2 out of 3 value chain steps for the automobile sector; **** SDGs with a high impact for the automobile and components industry.

## 4. Discussion

The developed method is used to select relevant environmental SDG indicators for the automobile industry. The method is not limited to the present design, but can also be broadened to other economic sectors and sustainability elements (society and economy). Companies can thereby select influenceable SDG indicators and, thus, contribute to the fulfillment of the SDGs. Therefore, the present paper goes beyond the existing studies [12,27,30] by selecting the influenced SDGs on the indicator level and by reducing the set of indicators by identifying impacts. Additionally, impacts of the automobile or transport sector on the selected indicators could be empirically proven in the present paper, as shown in Tables 3 and 4. Impacts were found for 29 of the 31 selected indicators, some of them in the double-digit percentage range. On indicator 3.9.3 (unintentional poisoning), no impact could be detected, since causes other than the automobile industry are responsible for a change in this indicator. Although the automobile industry could theoretically influence CO emissions (and CO poisoning is considered in the indicator), no causal relationship was identified, because unintentional poisoning by CO occurs in enclosed spaces and usually improper use or poor condition of indoor combustion devices and structural fires are the cause [71]. Additionally, no impact could be determined for indicator 12.5.1 (recycling rate), as no agreed calculation methodology is yet available for this indicator (Tier III). In the following, the results of this paper will be interpreted and compared with other studies. Further,

the methods, possible changes, and limitations are discussed. Finally, the future research potential is presented.

*4.1. Interpretation of the Results*

To select relevant environmental SDG targets for companies, the number of targets in [30] is reduced in several steps. First, this was done via exclusion criteria—SDG targets with letters and those that are mainly aimed at governmental action. Then, multiple and similar targets were summarized. Summarizing and thus reformulating similar indicators is deceptive, however, because it reduces the number but does not unselect any topics—the affected targets in the SDG framework remain the same. In addition, summarizing targets creates the problem that contributions can no longer be exactly matched with the SDGs. Compared to this, the present paper sets a narrower focus for the selection of relevant indicators. On the one hand, with the environment, one element of sustainability is considered, which is further limited by direct impacts. On the other hand, impacts for one particular industry are examined. This leads to the fact that, although the number of indicators is higher than the number of targets, the topics are selected more effectively. The indicators are reduced without being summarized or reformulated. This ensures an exact attribution to the SDG structure and allows determining the contributions of companies to the SDGs.

Comparisons of the selection decisions of the three criteria can only be carried out at the overarching goal level, since, with the exception of environmental indicators defined by the UNEP [53], other studies provide results on this level only. Table 5 shows an overview of the categorization of SDGs attributed to the environment. In [11] and [36] SDGs are categorized by ecology, society, or economy on the goal level, whereas in [53] and in the present paper the categorization is conducted on the indicator level. It is noticeable that, in the evaluation at the indicator level, more SDGs were categorized as environmental than at the goal level. In the case of SDGs 1, 2, 3, 8, and 9, individual indicators with direct environmental impact are identified, whereby the majority of the indicators of these goals have no (direct) environmental impact. Analyses at the goal level would filter out these indicators and their topics and thus not include them in the results, whereas analyses at the indicator level also consider SDGs that have another focus (such as social or economic) if at least one indicator also has an environmental impact. In this way, topics and impacts are not excluded. The results of the analyses at the indicator level are therefore similar. The main difference between the results of the first step in this paper (environmental impact, 124 indicators, 17 SDGs) and UNEP [53] (93 indicators, 16 SDGs) relates to the definition of environment. In the present paper, human health is considered as part of the environment, while basic services and financing issues (including for environmental protection) are excluded. In [53], it is done vice versa. In addition, in the present paper the current SDG indicators [4] with the latest changes are examined, while the other study uses a previous indicator set. The second step in this paper (direct environmental impact) reduces the number of selected indicators to 45, belonging to 12 SDGs.

The impact of the automobile industry (third step) on the SDGs is compared with three studies —see Table 6. The comparability with the studies mentioned above decreases if it is taken into account that, in the present paper, before assessing the impact of the automobile industry numerous indicators were already excluded in the first two steps. In this paper, the impact of the automobile industry on indicators with a direct environmental impact is evaluated. All the more interesting is the result that, in this paper, indicators which are influenced by the automobile industry are also identified for the mentioned SDGs of the three studies—except SDGs 5 and 17. It can be concluded from this that the automobile industry has a broad impact on the environment, and that this impact is apparent even in analyses without an ecological focus.

Definitions from van Zanten and van Tulder [30] were used to classify the ethical duties. The evaluation of the indicators in the present paper, for which the respective targets were also included, is therefore consistent with the evaluation of the targets from [30]. As shown in Tables 3 and 4, 28 of the relevant environmental indicators for the automobile industry deal with negative impacts, while three

have positive impacts. This strong imbalance for avoiding harm indicators can be explained by the focus on environmental protection. Environmental protection is mainly aimed at reducing damage. This relationship is also shown in [11]: for SDGs with environmental impacts, the reduction in negative impacts is in the foreground, whereas for social and economic SDGs the creation of positive impacts is predominant. Besides this, by definition, doing good is intended to create benefits that are perceived as additional. Since, in the present work, indicators have been evaluated that are influenced by automobiles, it is expected that the automobile industry will also take care on the negative impacts of these indicators. For the automobile industry, however, the selection of largely avoiding harm indicators may provide opportunities. According to [30], companies that are committed to the SDGs tend to select avoiding harm indicators, as the responsibility is more evident in these indicators and companies can thus contribute to the SDGs without being dependent on others. Nevertheless, it is also noted that companies will have to change their practices and also work on doing good indicators in order to achieve the SDGs by 2030.

*4.2. Discussion of the Methods*

The three criteria of environmental impact, direct impact, and automobile impact are used to identify the environmental indicators relevant for the automobile industry, in the sense of being influenceable [12,27]. The first two criteria are independent of industry and are used to select the SDG indicators with a direct environmental impact. The third criterion is used to evaluate the influence of an industry—in this case, the automobile industry—on indicators.

The first criterion, environmental impact, was determined on the basis of the sustainability hierarchy and the need for more environmental protection. The number of indicators selected in this stage demonstrates that environmental impact is a useful criterion for focusing the analysis on relevant indicators. On the one hand, this criterion is effective in filtering out about half of the 247 SDG indicators. On the other hand, due to the comprehensive definition of the environment, the group of 124 indicators selected is not too specific, but ensures the coverage of a wide range of topics. In this step, mainly indicators with social and economic content were filtered out. All the indicators with a financial content were assessed as economic, even if they collect financial resources for environmental protection (e.g., 1.a.2, 6.a.1 and 15.a.1). Indicators that promote sustainable development were evaluated as having an environmental impact, as this is part of sustainable development (e.g., 12.1.1 and 17.16.1). Basic services were assessed as social even if they also have potential environmental impacts (e.g., 7.1.1). Health issues were not counted as basic services, as they are part of the environmental definition in this paper and are therefore further investigated in the next steps. If an indicator includes both basic services and environmental topics, it has been evaluated as having an environmental impact (e.g., indicator 6.3.1).

In order to select influenceable indicators from the group of indicators with an environmental impact, direct impact was used as the second criterion. In this step, indicators are selected whose environmental impact does not depend on other activities. Thus, contributions to these indicators also directly contribute to environmental protection. In this way, the group selected in the first step was reduced to 45 indicators. Environmental conditions are considered as indicators with direct impact (e.g., 2.5.2 and 6.3.2) and are selected. All the preparatory measures such as plans, strategies, or political frameworks are filtered out. Such structures are necessary conditions for environmental protection. However, they are a preparatory stage, and actual protection must be performed afterwards. (e.g., 1.5.3, 6.5.1, and 14.2.1). In addition, indicators that include general health and are not caused by ecosystems or resources are filtered out in this step (e.g., 2.2.2., 3.3.3, and 4.2.1.). A possible improvement of the methods would therefore be to filter out general health indicators in the first step, since it is a basic service anyway. This would reduce the number of selected indicators in the first step, but would lead to the same results from the second step onwards.

The automobile is used to select indicators that are influenced by the automobile industry. In this way, 31 relevant environmental indicators are identified, 27 of which are independent. These indicators



belong to 12 SDGs and thus cover a wide range of environmental issues and demonstrate the diverse influence of the automobile on the SDGs. The topics not influenced by the product include nature conservation zones (e.g., 14.5.1 and 15.4.1) and the installed renewable energy-generating capacity in developing countries (e.g., 7.b.1 and 12.a.1). By using the main product of the investigated industry as an impact, the main activity is covered and the influence can be defined well by inputs and outputs. For industries with different products, the totals of inputs and outputs can also be used as an impact. The definition of impact of an industry could also be broadened—e.g., the pure possibility to exert influence, or additional engagement independent of the own value chain. This would more realistically represent the actual impact and also increase the sphere of influence. In large companies, for example, additional activities can also be carried out, such as providing food for employees, supplying energy and water to the region, or other extraordinary commitments, especially in crisis periods. However, an extension of the impact definition would also bring with it the challenge of defining limits to what and how much of the potentially influenceable issues can ultimately be considered as "impact". Assuming the impact of the automobile industry through the product also falls short when considering automobile companies as mobility service providers. Moreover, such a definition primarily applies to automobile manufacturers. For other participants in the automobile industry, such as suppliers, it would have to be investigated which sub-areas they influence. Ultimately, the binary classification of whether an influence exists bears the risk that influences of different magnitudes are treated equally, since influences are considered independently of relative contribution to an environmental problem. Therefore, the level of influence would have to be calculated additionally.

The analysis of the empirical impact demonstrates that, by applying the present method, indicators can be selected on which the automobile industry has an impact in practice. It should be noted that the percentages given in Tables 5 and 6 do not necessarily reflect the share of the automobile industry in an indicator. On the one hand, because no complete analysis of all possible impacts was carried out, one example was already sufficient to demonstrate an impact. On the other hand, the evaluation frames are not consistent for every indicator. For example, the transport sector contains other parts besides road transport. Road transport itself can be differentiated into further parts—e.g., passenger cars, light commercial vehicles, and trucks. However, influences from the automotive industry and the transport sector were treated equally in the present paper. For this reason, the percentages between the indicators are not comparable. Furthermore, for some indicators, reasoned estimates had to be made if no dedicated data were available. In the case of indicator 3.9.3 (unintentional poisoning), no impact could be identified based on this. For indicator 8.4.2 (domestic material consumption (DMC)), the proportion of indicator 8.4.1 (material footprint) was used, since DMC by definition cannot be disaggregated to economic sectors. For indicator 15.4.2 (mountain green cover index), it is possible that part of the area is also used for the automotive industry—e.g., for cork oaks. However, since this could neither be substantiated with data nor disproved, the total economic share of wood used for automobiles (without differentiation for mountains) was assumed to be proportional for this indicator.

Besides, it does not necessarily mean that an indicator is controllable by the automobile industry if it is influenceable—e.g., for the indicators 1.5.1 (number of deaths, missing persons, and directly affected persons attributed to disasters per 100,000 population), 3.9.1 (mortality rate attributed to household and ambient air pollution), and 15.5.1 (Red List Index), the automobile industry causes inputs or outputs that influence the indicator. For the three examples mentioned, $CO_2$ emissions lead to an increase in the average temperature, particulate emissions lead to an increase in particle concentration, and land use and land change lead to habitat destruction. However, the impact (indicator change) is not linear and not exclusively attributable to the activities of the automobile industry. The impact depends on the actual situation and other stakeholders. This does not mean that the automobile industry should not work on indicators that can be influenced, as it still can control inputs and outputs of the indicators selected in the present paper. However, it should be considered that the automobile industry cannot achieve these targets and goals on its own. Partnership from SDG 17 becomes the focus of attention at this point.

A further limitation is that, since one example per stage is sufficient for the selection decision, Table S1 in the Supplementary Materials does not contain a complete analysis of the impacts per indicator. If a topic has multiple impacts—for example, climate change impacts human health as well as ecosystems—usually only one impact is used for the explanation of the decision. For this reason, no analysis of the most frequent environmental impacts can be made. In addition, no metadata are available for some indicators—marked "Tier III" in Table S1. As such indicators are assessed by name, there is uncertainty as to whether the assessment will remain valid once the methodology is developed and metadata are published. With 12.5.1, one of the environmental indicators relevant for the automobile industry belongs to this group with uncertainty.

*4.3. Future Research Potential*

The results of the present paper serve as a basis for further work. The qualitatively determined, relevant environmental SDG indicators for the automobile industry indicate which indicators are influenceable. The impact was additionally empirically proven. Quantitative analyses are needed to determine the level of influence consistently for all indicators. Therefore, a calculation model can be developed that quantifies the impacts (see Table 5, Table 6, Table S1, Table S2, and Table S3 for impact examples) of the automobile industry within defined system boundaries. The impacts can then be calculated for indicators as proportions for a country or region. In this way, the contributions of an industry or a company to the SDGs can be determined, thus solving the problem of the lack of methods for evaluating contributions. The procedure described may be necessary and logically plausible, but obstacles still have to be overcome in future research. For example, in addition to indicators where the proportions are obvious (e.g., 13.2.2 "Total greenhouse gas emissions per year"), there are also indicators where further work still needs to be undertaken. In order to be able to quantify the impact on indicator 1.5.1 "Red List Index", for example, it is first necessary to determine which aspects of the automobile industry influence this indicator. Subsequently, data must also be collected for the determined influencing variables until they can be calculated to a proportion. Another promising research area is the prioritization of these relevant indicators through a multi-criteria analysis. In addition to the quantified influence, other criteria, such as those defined by Allen et al. [105]—that is, level of urgency, systemic impact, and policy gap—can be used for this purpose.

## 5. Conclusions and Outlook

To support the private sector in meeting the SDGs, a method for selecting relevant environmental SDG indicators for the automobile industry was developed in this paper. Therefore, the three criteria—environmental impact, direct impact, and automobile impact—were defined, by which 31 relevant out of the current 247 SDG indicators were selected. These indicators belong to 12 SDGs, so that more than two thirds of the SDGs are influenced by the automobile industry. SDG 6 (clean water and sanitation) and SDG 15 (life on land), each with five indicators, contain the most, followed by SDG 12 (responsible consumption and production) with four indicators. For the selected indicators, empirical evidence of the impact of the automobile industry was additionally examined and found for 29 of the 31 indicators. This analysis demonstrates that, by applying the present method, indicators can be identified on which the automobile industry has an impact in practice. In order to make its ecological contribution to the SDGs, the automobile industry should work on the indicators selected and optimize its influences along the value chain accordingly. The vast majority of the indicators selected here measure negative impacts and are concerned with avoiding harm. This offers the advantage that the responsibility is more recognizable and individual companies can contribute to environmental protection.

The reason for the relatively high number of influenced SDGs is the analysis at the indicator level. In this way, it is possible to select influenceable indicators from one SDG even if this SDG has a different focus, as is the case for environmental impacts in SDGs 1, 8, and 9. Analyses at goal level would filter out these indicators and their topics and thus not include them in the results. The impact

analysis at the indicator level is also necessary to quantify contributions to the fulfillment of the SDGs, since indicators substantialize and operationalize targets. For the attribution of contributions to the SDGs, it is particularly important that the original SDG framework is not modified by reformulation or summarization.

The results of this paper serve as a basis for further work. Thus, in the next step, the level of influence for the relevant environmental indicators can be quantified, as further described in the discussion chapter. In addition, other criteria can be examined and then compared with the presented results. The present method can be broadened to include the other elements of sustainability (society and economy), as well as other types of impact (e.g., indirect). Finally, it is also interesting to use this criteria-based approach to determine the relevant indicators of companies, other industries, or economic sectors. This makes it possible to determine who can contribute to which indicators. Additionally, such comparisons clarify for which indicators there are few contributions. Such indicators, which are influenced to a small extent by the private sector, must then be addressed by, or with, other stakeholders in order to meet the SDGs by 2030.

**Supplementary Materials:** The following are available online at http://www.mdpi.com/2071-1050/12/21/8811/s1: Table S1: Selection of all 247 indicators with explanation. Table S2: Empirical evidence of the impact—avoiding harm indicators. Table S3: Empirical evidence of the impact—doing good indicators.

**Author Contributions:** Conceptualization, S.L., M.B., J.C., K.M.-R., G.B., and M.F.; methodology, S.L., M.B., J.C., K.M.-R., and G.B.; validation, J.C. and M.B.; formal analysis, S.L. and J.C.; investigation, S.L.; writing—original draft preparation, S.L.; writing—review and editing, S.L., M.B., J.C., K.M.-R., and M.F.; visualization, S.L.; supervision, M.B., K.M.-R., G.B., and M.F. All authors have read and agreed to the published version of the manuscript.

**Funding:** This research received no external funding.

**Acknowledgments:** The authors would like to thank Lina Kindermann and James Magness for very helpful discussions and comments on earlier versions of this paper.

**Conflicts of Interest:** The authors declare no conflict of interest. The results, opinions and conclusions expressed in this paper are not necessarily those of Volkswagen Aktiengesellschaft.

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
