# Peer review of "Criteria-Based Approach to Select Relevant Environmental SDG Indicators for the Automobile Industry"

_sustainability, doi:10.3390/su12218811_

Round 1
Reviewer 1 Report
Dear Authors,
The correct link must be correctly reflected in "Error! Reference source not found"(Lines 314, 444).
All clarifications and changes made provide adequate response to the comments made.
Congratulations on the paper done.
Kind regards
Reviewer 2 Report
The version resubmitted has been considerably strengthened and provides a more solid approach than the former submission. Authors have carried out a deeper review of scholar literature and made great efforts to provide empirical evidence of the impact of the automobile sector on the selected indicators. That reduces the sound of potential bias and deserves congratulations.
I will, however, point out some little issues that in my own view should still be addressed to make the most of the research and get it published:
- In its current version, the comparison of the results with other studies has increased its methodological importance… to the point that now it appears three times in the titles of the sections (2.4; 3.3; 4.1). That might be confusing. I find that a reformulation is needed, in order to clarify from the very beginning what exactly is going to be compared in each subsection.
- I have some concerns about citation n. 11, especially when it is attributed to EAS without further explanation. Sometimes, the source cited as 11 appears in the text as a general support reference, which is OK (in lines 97, 134, 188, 342 and 374). However, in line 225 there is an automatic correlation between the authors of that paper (Beck and Buddmeier) and EAS as an institution. Several times afterwards, the findings of those authors are directly attributed to that EAS (in lines 229, 232 or 235, and what is more important, in tables 5 and 6). As the link provided in the references section for the paper of Bech and Buddmeier has not given me direct access to it, I am not able to test if those authors work does indeed reflect the EAS view. By the way, it is not so obvious what that acronym stands for.
- There is some format problem regarding inside and crossing references to tables and figures, as a systematic error message appears when mentioning them in the text (lines 314, 445-446). Something similar happens in line 285 (reference to Table 5 is duplicated) and 296 (there is a tab missing after mentioning Table 6). The missing tab is also present in line 235
Author Response
Please see the attachment.

This manuscript is a resubmission of an earlier submission. The following is a list of the peer review reports and author responses from that submission.
Round 1
Reviewer 1 Report
The paper focuses at how private sector (automobile industry in particular) may contribute to SDG achievement and, in order to ease the task, it tries to filter the goals and select the most relevant indicators. It is a very interesting and new point of view, something that deserves to be underlined and welcome.
However, the method followed to reach a conclusion (that is, the layout of a set of indicators on which automobile industry should concentrate their efforts regarding SDG challenge) is strictly theoretical and argumentative. Information included in the supplementary materials provides justification of each decision (include or exclude an indicator for the final list) but no deep/critical explanation. The feeling of the reader is that the table in those materials is just built to back up authors preconceptions and although argumentation is most of the time quite convincing, a shadow of bias remains. In fact, the authors seem to recognize that weakness of the approach, especially in 394-402.
I will provide only an example to enlighten my appreciations: indicators 15.1.1 and 15.2.1 are finally selected as relevant because automobile production requires "natural rubber and cork". Nevertheless, forest is obviously composed by many other kind of trees, so the relationship between those trees comsumption by automobile companies and the "forest area as a proportion of total land area" (15.1.1) is not so clear. What is more, how that comsumption influences "sustainable forest management" (15.2.1) is, if possible, much more controversial as the management of the forest is somebody else's responsibility and it is not easy to see how automobile companies may influence forest managers.
Chapter 4.2 of the main text is honestly written, admitting its limitations. Having this in mind, I would still suggest authors to make an additional effort to strengthen their choices of relevant indicators with a more solid approach. That would add academic value to the paper and improve its profit for automobile companies, who are the final recipients of this piece of work. In order to rethink the approach, I would also suggest to define more precisely the target in order to determine whether we are in search of relevant indicators regarding automobile production (mobility as an industry) or automobile use (mobility as a product); that might be helpful to get more robust conclusions.
Finally, there is some format problem regarding inside and crossing references to tables and figures as a systematic error message appears when mentioning them in the text.
Good luck!
Reviewer 2 Report
Dear Authors,
In this paper, a criteria-based approach to select relevant environmental Sustainable Development Goals indicators for the automobile industry is developed. The three criteria environmental impact, direct impact and automobile impact are defined.
The paper deals with a topic of interest but has some issues that need to be reviewed before publication. The following are the aspects that need to be improved:
- The way to reflect some quotations and incorporate them in the text taking into account the rules of style of the journal (for example, in line 43: "As the authors of [5] point out,..." should be expressed as "Stafford-Smith et al. [5] point out,..."; also in line 92 "The need for methods to evaluate contributions to the SDGs is also described in [29]"...)
- The correct link (Table, Figure,...) must be correctly reflected in "Error! Reference source not found".
- A series of causal relationships are established that present a certain degree of subjectivity and are not empirically demonstrated. The robustness of the proposed model has not been verified. Therefore, tests must be carried out to test the utility and viability of the model. The results obtained in these tests must be able to corroborate the capacity of the model to quantify the impacts of the automotive industry in a given area (several countries or a specific region such as the European Union).
Kind regards,